# DiffLM: Controllable Synthetic Data Generation via Diffusion Language Models

## Abstract

Recent advancements in large language models (LLMs) have significantly enhanced their knowledge and generative capabilities, leading to a surge of interest in leveraging LLMs for high-quality data synthesis. However, synthetic data generation via prompting LLMs remains challenging due to LLMs' limited understanding of target data distributions and the complexity of prompt engineering, especially for structured formatted data. To address these issues, we introduce DiffLM, a controllable data synthesis framework based on variational autoencoder (VAE), which further (1) leverages diffusion models to reserve more information of original distribution and format structure in the learned latent distribution and (2) decouples the learning of target distribution knowledge from the LLM's generative objectives via a plug-and-play latent feature injection module. As we observed significant discrepancies between the VAE's latent representations and the real data distribution, the latent diffusion module is introduced into our framework to learn a fully expressive latent distribution. Evaluations on seven real-world datasets with structured formatted data (i.e., Tabular, Code and Tool data) demonstrate that DiffLM generates high-quality data, with performance on downstream tasks surpassing that of real data by 2%–7% in certain cases. Data and code will be released upon acceptance.

## 1 Introduction

Data Synthesis has become an indispensable technique in current machine learning research, enabling rapid generation and modification of datasets (Bauer et al., 2024), allowing researchers to experiment with various scenarios and model architectures without the extensive processes associated with real-world data collection. Meanwhile, with the rapid advancements in large language models (LLMs), recent research in natural language processing (NLP) has increasingly focused on leveraging LLMs for synthetic data generation. Early efforts attempted to fine-tune LLMs to align with real data distributions (Keskar et al., 2019; Anaby-Tavor et al., 2020; Borisov et al., 2023). As the in-context learning capabilities of LLMs have improved, some studies have explored zero-shot or few-shot prompting of LLMs to generate synthetic data (Ye et al., 2022a; Wei et al., 2024).

Despite the progress achieved, generating high-quality synthetic textual data using LLMs remains challenging, particularly for structured data (Josifoski et al., 2023; Li et al., 2022). First, LLMs often lack a global understanding of the target data distribution when generating synthetic data. Even after fine-tuning, it is difficult to inject information about complex and varied distributions into current LLM architectures, often resulting in outputs with low diversity and instances of data copying (Wu et al., 2024; Yu et al., 2023). Moreover, existing LLM-based synthetic data generation methods typically involve complex pipelines and post-processing mechanisms, such as prompt engineering, multi-agent frameworks, and iterative sampling (Dekoninck et al., 2024; Wu et al., 2024). These complexities hinder the rapid adaptation of LLMs to new tasks, limiting their utility in dynamic research and industrial scenario. Concurrently, the remarkable performance of variational autoencoders (VAEs) and diffusion models in image synthesis tasks (Betker et al., 2023; Rombach et al., 2022) has spurred interest in adapting these techniques to other modalities (Borisov et al., 2023; Li et al., 2022; Gong et al., 2023). Although some works have introduced latent spaces into language models for simple tasks like style transfer or topic generation (Yang & Klein, 2021; Li et al., 2022), our preliminary experiments indicate that directly applying the latent distributions learned by VAEs often results in outputs that are unrelated to the real data. Similar issues also have been addressed

in prior works (Amani et al., 2024; Havrylov & Titov, 2020; Bowman et al., 2016). This challenges the direct application of these methods in more complex scenarios for synthetic data generation.

To address these challenges, we propose DiffLM, a novel framework that leverages a plug-and-play latent space to provide data distribution information for LLMs during data generation. First, to decouple the learning of real data distributions from the LLM's training objectives, we develop a latent space using a VAE model to capture external information, mapping samples from the real dataset to latent vectors. However, we observed that sampling points from a Gaussian distribution obtained from naive VAE that cannot generate realistic results. To overcome the poor quality of data generated by sampling from VAE, we employ a latent diffusion method that linearly adds noise to the latent space over time. A denoising network is then trained to learn these noises in the reverse process, reducing efficiency loss in data synthesis due to sampling failures. Finally, we design a soft prompting method to inject latent features into the LLM decoding process, resulting in controllable, high-quality synthetic data. We evaluate our method on seven real-world structured formatted datasets, ranging from relatively simple table synthesis to more complex code and tool synthesis tasks. Experiments demonstrate that DiffLM can generate high-quality results, and ablation studies confirm the effectiveness of each component in our proposed method.

The contributions of this paper are threefold:

- **Decoupling Data Distribution Learning**: We proposed a new VAE-based LLM framework for data systhesis, which decouples the learning of real data distribution information from the training objectives of the LLM by introducing the a small projection network.

- **High-Quality Synthetic Data Generation**: Based on our observations, the meticulously designed VAE and diffusion structures effectively model the distribution of real data, enabling the generation of high-quality synthetic data. In all tasks, the quality of the generated data is comparable to or even surpasses that of the real data.

- **Comprehensive Evaluation**: We validate the high quality of data generated by DiffLM across three distinct scenarios and seven datasets, underscoring its robustness and adaptability in advancing synthetic data generation for natural language processing.

## 2 RELATED WORKS

**Large Language Models in Data Synthesis.** The recent advancement in the generative capabilities of LLMs has motivated numerous exploratory works aiming to leverage these models for data augmentation in areas such as text classification (Ye et al., 2022a; Li et al., 2023), information extraction (Tang et al., 2023; Josifoski et al., 2023), and tabular data generation (Borisov et al., 2023; Xu et al., 2024). A comprehensive survey conducted by Long et al. (2024) proposes a prompt-based generic workflow for synthetic data generation, curation, and evaluation. And multiple advanced works have attempted to fine-tune language models for data synthesis in recent years (Anaby-Tavor et al., 2020; Kumar et al., 2020; Dinh et al., 2022; Borisov et al., 2023; Xu et al., 2024). Specifically, these methods involve fine-tuning LLMs on a small amount of gold data for language modeling, followed by the use of various sampling methods to generate data. However, a major challenge remains in ensuring that synthetic data accurately reflects real-world distributions. Veselovsky et al. (2023) has shown that LLM-generated data can sometimes diverge from actual data distributions, leading to unfaithful representations that may hinder model training. Some studies have explored data selection (Puri et al., 2020) or data augmentation (Ye et al., 2022b) to address this distribution gap, but there remains significant room for improvement.

**Latent Variable Models in Text Generation.** Latent variable models have made significant advances in computer vision in recent years (Yu et al., 2022a; Gu et al., 2022; Luo et al., 2023a; Gulrajani et al., 2017), achieving high-quality generation results, flexibility and effectiveness, as well as robustness to noise perturbations. In particular, latent diffusion models, such as DALL-E (Betker et al., 2023) and Stable Diffusion (Rombach et al., 2022), operate their diffusion processes in a latent space rather than directly in data space, enabling a near-optimal balance between generation quality and computational efficiency. In text generation, several works (Bowman et al., 2016; Wiseman et al., 2018; Kaiser & Bengio, 2018; Havrylov & Titov, 2020; Ding & Gimpel, 2019; Li et al., 2022; Gu et al., 2023; Borisov et al., 2023; Amani et al., 2024) have attempted to combine latent

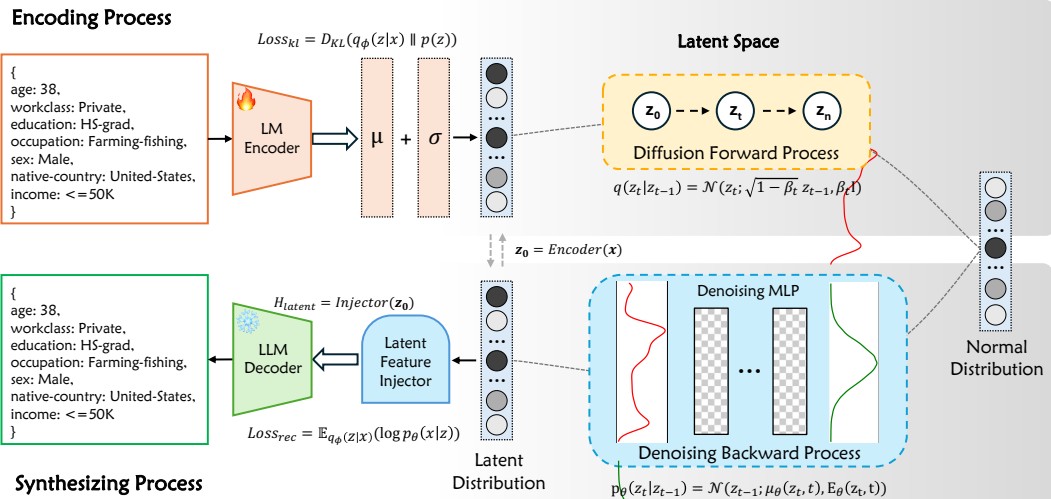

Figure 1: Overview of our **DiffLM**. The trainable lanaguage model (LM) works as VAE encoder while the fixed LLM decoder serves as VAE decoder. We further (1) introduced a Diffusion module to learn the latent space, and (2) employ a latent feature injector with soft prompting to align latent vector space with LLM decoder.

spaces with language models to accomplish tasks such as sentence representation, text style transfer, and dataset augmentation. Additionally, some studies have explored the use of diffusion for plug-and-play controllable generation (Li et al., 2022; Gong et al., 2023), aiming to steer the outputs of pre-trained language model using auxiliary modules. While these works share a similar perspective with ours, we tackle a more challenging scenario of structured data synthesis and thoroughly investigate multiple methods of latent knowledge injection. To the best of our knowledge, our work is the first to combine VAEs and denoising diffusion models with large language models for high-quality data synthesis.

## 3 METHODOLOGY

Figure 1 illustrates the main pipeline of our proposed DiffLM. First, we define an encoder to map discrete text into a continuous latent space (Section 3.2). Second, although the features of the data are extracted and compressed, conventional latent embeddings in text VAEs often lead to decoding failures due to underutilized or empty regions in the latent space. To address this issue, we train a diffusion model on the latent space (Section 3.3). Finally, to incorporate encoded prior knowledge into the decoding stage of large language models, we propose a novel soft prompt injection method to steer the decoding process (Section 3.4).

### 3.1 PROBLEM FORMULATION

We begin by defining $\mathcal{D}$ as a known small set of real-world distribution data, where each element $x$ represents a real sample. We define $G$ as the synthetic data generator, which learns the distribution of $\mathcal{D}$ and generates a set of synthetic samples, $\mathcal{D}_{syn}$, ensuring that the model does not simply memorize and reproduce the same real samples, meaning $\mathcal{D} \cap \mathcal{D}_{syn} = \varnothing$. It should be noted that we focus on the task of unconditional data synthesising using LLMs, where $G$ generates synthetic samples independently of any additional context, i.e., without using explicit prompt text.

### 3.2 VAE-BASED REPRESENTATION LEARNING

**Feature Encoding:** In standard VAEs, an encoder is typically employed to map input data into a latent space. Given structured text data $s_i$, we utilize a learnable Transformer-based pre-trained language model (Vaswani et al., 2017; Devlin et al., 2019; Raffel et al., 2020) to obtain the representation vector $x_i \in \mathbb{R}^{d \times 2}$, which can be split into the mean and variance. Using the re-parameterization

trick (Kingma & Welling, 2014), we then obtain the latent feature $z \in \mathbb{R}^d$:

$$z = \mu + \sigma \odot \epsilon, \tag{1}$$

where $\mu$ and $\sigma$ are the mean and standard deviation output by the encoder, and $\epsilon$ is sampled from a standard normal distribution $\mathcal{N}(0, I)$.

**LLM Decoding:** After generating the latent feature $z$, we employ a frozen-parameter LLM to reconstruct the input text $s$ in a causal language modeling manner. The rationale for freezing the LLM parameters is to avoid retraining and to preserve its general knowledge and reasoning capabilities. Consequently, aligning the two different modalities, whereas the latent space and the LLM input space, presents a significant challenge. To address this, we propose a novel latent feature injector using soft prompting and design a corresponding injector network; specific details are provided in Section 3.4.

**VAE Training Objective:** The VAE model is typically trained using the Evidence Lower Bound (ELBO) loss function. Following previous work (Burgess et al., 2018), we adopt the $\beta$-VAE training strategy (Higgins et al., 2017), which introduces a weighting parameter $\beta$ to control the contribution of the KL divergence loss in the total loss function. Specifically, when $\beta = 0$, the model reduces to a standard autoencoder. For $\beta > 0$, the KL constraint encourages learning a smoother latent space:

$$\text{ELBO}_\beta = L_{rec} - \beta L_{kl}, \tag{2}$$

$$L_{rec} = \mathbb{E}_{q_\phi(z|x)}\big(\log p_\theta(x|z)\big), \tag{3}$$

$$L_{kl} = D_{\text{KL}}\big(q_\phi(z|x) \parallel p(z)\big), \tag{4}$$

where $p_\theta(x|z)$ is the language modeling reconstruction likelihood, $q_\phi(z|x)$ is the approximate posterior, and $p(z)$ is the prior over the latent space, i.e., Gaussian distribution. In our model design, considering the denoising network of latent diffusion, we adopt an decreasing $\beta$ adjustment strategy. We initially set a larger $\beta$ weight to enforce a strong regularization on the latent space. As the reconstruction loss convergence slows, we decrease the $\beta$ value to allow the model to focus more on reconstruction accuracy. Additionally, we employ an early stopping mechanism to prevent overfitting.

### 3.3 LATENT SPACE DENOISING

Although VAE can learns latent space representations of data, directly sampling from the prior distribution $p(z)$ often exhibit low quality generated samples. In our preliminary experiments, we observed that directly utilizing the latent features learned by the VAE frequently produces text that is unrelated to the target data distribution. This issue arises due to the discrepancy between the encoder's learned posterior distribution $q_\phi(z|x)$ and the prior $p(z)$. To address this problem, we introduce a diffusion model in the latent space to more accurately model the true distribution of the latent features. Inspired by Zhang et al. (2024), we extract the latent vectors $z \in \mathcal{Z}$ from the trained VAE for each data point $x \in \mathcal{D}_{\text{train}}$. Starting from the initial latent vector $z_0$, we progressively add noise over time following a linear schedule to get $z_t$. During the reverse diffusion process, we employ a standard continuous denoising network to recover $z_0$ (Song et al., 2021). For the training objective, we optimize the diffusion model through denoising score matching (Karras et al., 2022):

$$z_t = z_0 + \sigma(t)\epsilon, \epsilon \in \mathcal{N}(0, I), \tag{5}$$

$$dz_t = -\dot{\sigma}(t)\sigma(t)\nabla_{z_t}\log p(z_t)dt + \sqrt{2\dot{\sigma}(t)\sigma(t)}d\omega_t, \tag{6}$$

$$\mathcal{L}_{\text{diff}} = \mathbb{E}_{t\sim p(t),\, z_0\sim p(z_0),\, \epsilon\sim\mathcal{N}(0,I)} \left\| \epsilon_\theta(z_t, t) - \epsilon \right\|^2, \tag{7}$$

In forward process Eq.5, $z_t$ is the latent variable at time $t$, and $\sigma(t)$ is a time-dependent noise scale function. As for backward process Eq.6, $\dot{\sigma}(t)$ stands for the time derivative of $\sigma(t)$, and $\nabla_{z_t}\log p(z_t)$ is the gradient of the log probability density with respect to $z_t$, also known as the score function, and $d\omega_t$ is an increment of the Wiener process (standard Brownian motion). For diffusion model training loss Eq.7, $\epsilon_\theta(z_t, t)$ is the neural network that predicts the noise $\epsilon$ given $z_t$ and $t$. The detailed description for diffusion model could be found in Appendix A.1.

### 3.4 LATENT FEATURE INJECTION

After constructing a latent space that captures the true data distribution, two challenges remain: 1) *Aligning latent space with LLM's input space.* How can the decoding LLM process the latent vector

Table 1: Performance of downstream tasks using generated **tabular** data. We evaluate the quality from: performance in machine learning efficiency (**MLE**) task, and column-wise distribution density estimation ($\rho$) task. $\uparrow, \downarrow$ indicate that higher (or lower) metrics correspond to better performance. **Boldface** indicates DiffLM surpasses the SoTA model based on language models. **Red Boldface** denotes DiffLM exceeds the MLE performance achieved using real data.

| Method | Adult | | Default | | Magic | | Shoppers | | Beijing | |
|---|---|---|---|---|---|---|---|---|---|---|
| | MLE $\uparrow$ | $\rho \downarrow$ | MLE $\uparrow$ | $\rho \downarrow$ | MLE $\uparrow$ | $\rho \downarrow$ | MLE $\uparrow$ | $\rho \downarrow$ | MLE $\downarrow$ | $\rho \downarrow$ |
| Real | 0.927 | - | 0.770 | - | 0.946 | - | 0.926 | - | 0.423 | - |
| SMOTE | 0.899 | 1.60 | 0.741 | 1.48 | 0.934 | 0.91 | 0.911 | 2.68 | 0.593 | 1.85 |
| CTGAN | 0.886 | 16.84 | 0.696 | 16.83 | 0.855 | 9.810 | 0.875 | 21.15 | 0.902 | 21.39 |
| TVAE | 0.878 | 14.22 | 0.724 | 10.17 | 0.887 | 8.250 | 0.871 | 24.51 | 0.770 | 19.16 |
| GOGGLE | 0.778 | 16.97 | 0.584 | 17.02 | 0.654 | 1.900 | 0.658 | 22.33 | 1.090 | 16.93 |
| CoDi | 0.871 | 21.38 | 0.525 | 15.77 | 0.932 | 11.56 | 0.865 | 31.84 | 0.818 | 16.94 |
| TabSyn | 0.915 | 0.58 | 0.764 | 0.85 | 0.938 | 0.88 | 0.920 | 1.43 | 0.582 | 1.12 |
| GReaT | 0.913 | 12.12 | 0.755 | 19.94 | 0.888 | 16.16 | 0.902 | 14.51 | 0.653 | 8.25 |
| DiffLM | 0.894 | **9.16** | **0.793** | **9.33** | **0.910** | **7.04** | **0.912** | **14.43** | 0.717 | **6.05** |

$z$ to steer a powerful language model for realistic data generation? 2) *Seamless Integration with LLM Knowledge*: How can we integrate external information without disrupting the LLM's internal knowledge? Motivated by adapter training methods in LLM fine-tuning (Lester et al., 2021; Li & Liang, 2021; Houlsby et al., 2019; Liu et al., 2023a), we consider the soft prompt latent injection approach to incorporate $z$ into LLM decoding without training the model weights. Specifically, after obtaining the latent representation $z$, we use an upper MLP to map it into $k$ soft prompt token embeddings, denoted as $\mathbf{H}_{\text{latent}} \in \mathbb{R}^{k \times d}$. These soft embeddings serve as a steering vector, which is concatenated before the <BOS> token to assist the LLM in decoding. The detailed process is illustrated in Figure 4. We also conduct ablation experiments in Section 5.1 with the other two injection methods proposed by Li et al. (2020), which validated that our methods obtain the best reconstruction loss and downstream task performance.

## 4 EXPERIMENTS

In this section, we evaluate the generation quality of the DiffLM method on multiple public benchmarks across three tasks: 1) Tabular Data Generation: We compare DiffLM with SoTA tabular generation algorithms, demonstrating its strong capability in structured data generation. 2) Code Generation: DiffLM showcases the ability to integrate structured data priors with its internal knowledge. The results on synthetic data are even better than real ones. 3) Tool Generation: DiffLM can quickly adapt to complex function call scenarios, highlighting its flexibility and adaptability.

### 4.1 TABULAR DATA GENERATION

**Benchmarking.** We selected five publicly available datasets for evaluation, encompassing both classification and regression tasks: Adult, Beijing, Default, Magic, and Shoppers. The properties of datasets are presented in Table 5. To assess the quality of synthetic data, we employed two perspectives: 1) **Low-order statistical metrics**, where we quantified column-wise density estimation using the Kolmogorov-Smirnov Test for numerical columns and the Total Variation Distance for categorical columns; 2) **Downstream task performance**, where we measured the predictive accuracy on test data of classifiers or regressors trained on the generated data.

**Baselines.** We selected a comprehensive set of classic and SoTA tabular data generation models with diverse architectures for comparison. First, we consider the performance on real data as the upper bound for evaluation. Secondly, we included the classic method, synthetic minority oversampling technique (SMOTE) (Chawla et al., 2002), which generates new synthetic data patterns by performing linear interpolation between minority class samples and their $k$ nearest neighbors. Additionally, for neural network-based tabular generation algorithms, we considered six baseline

Table 2: pass@k scores on **HumanEval** and **MBPP**. We follow Chen et al. (2021) for estimating pass@k, where $n > k$ solutions are generated per problem with p = 0.95 and a temperature of 0.2 to calculate the success rate with zero-shot learning. **Boldface** indicates that DiffLM surpasses the performance achieved with real data. **Red Boldface** indicates that DiffLM surpasses the base model's performance.

| Model | Size | HumanEval | | | MBPP | | |
|---|---|---|---|---|---|---|---|
| | | pass@1 | pass@10 | pass@100 | pass@1 | pass@10 | pass@100 |
| GPT-4 | - | 67.00 | - | - | - | - | - |
| CodeLLaMA | 7B | 33.50 | 59.60 | 85.90 | 41.40* | 66.70* | 82.50* |
| | 34B | 48.80 | 76.80 | 93.00 | 55.00* | 76.20* | 86.60* |
| Mistral-Base | 7B | 27.79 | 41.22 | 56.37 | 37.31 | 52.02 | 59.65 |
| | 12B | 10.12$^\dagger$ | 20.91$^\dagger$ | 28.93$^\dagger$ | 43.38 | 61.44 | 69.09 |
| Mistral-Instruct | 7B | 36.09 | 52.95 | 64.18 | 38.45 | 50.77 | 59.17 |
| | 12B | 7.08$^\dagger$ | 12.43$^\dagger$ | 16.14$^\dagger$ | 52.20 | 63.61 | 69.02 |
| **Mistral-Real-Code** | 7B | 28.58 | 42.24 | 54.24 | 27.15 | 42.21 | 48.14 |
| | 12B | 36.97 | 52.04 | 60.95 | 34.79 | 45.49 | 50.22 |
| **Mistral-DiffLM-Code** | 7B | **35.37** | **47.36** | **54.38** | **32.70** | 41.65 | 47.39 |
| | 12B | **42.24** | **56.02** | **61.97** | **44.42** | 52.35 | 55.70 |

\* These results are evaluated under a 3-shot setting.

$^\dagger$ The vanilla Mistral-Nemo 12B models fail to pass the HumanEval benchmark, resulting in a lower score. We have conducted multiple evaluations and report the average performance.

models across different architectures: 1) *GAN*-based models: CTGAN (Xu et al., 2019); 2) *VAE*-based models: TVAE (Xu et al., 2019), GOGGLE (Liu et al., 2023b); 3) *Diffusion*-based models: Codi (Lee et al., 2023), TabSyn (Zhang et al., 2024); 4) *LLM*-based: GReaT (Borisov et al., 2023), which attempts to fine-tune a GPT-2 (Radford et al., 2019) for table synthesis. It is worth noting that we compare with the current strongest generative models not to merely outperform them in tabular generation but to demonstrate that our flexible DiffLM architecture can achieve comparable performance while offering additional advantages.

**Evaluation.** Table 1 presents the quality assessment results of the generated data. For different tabular datasets, we train a XGBoost classifier or a regressor using the synthetic data to predict the label column values, using AUC and RMSE to evaluate the accuracy, respectively. From the results, DiffLM outperforms the current language-model-based SoTA (GReaT model) on most datasets. Notably, on the *Default* dataset, the prediction accuracy using DiffLM's synthetic data surpasses that obtained by training on real data. This suggests that DiffLM's approach of integrating the real data distribution with its own learned knowledge can provide richer information for downstream tasks while preserving the original data structure. In other words, the synthetic data generated by DiffLM contains additional knowledge compared to real data, which is challenging to achieve with previous methods. Moreover, our generated results achieve performance comparable to prior methods in column-wise distribution density estimation. Although the TabSyn method attains superior performance on several datasets, it should be noted that our approach focuses on general, pluggable generation control for large language model, rather than training data synthesis models from scratch for specific domains. Despite this, in tabular data generation, our method's performance is on par with these domain-specific methods.

## 4.2 CODE GENERATION

**Benchmarking.** In the code generation scenario, to simplify the problem, we focus on Python code and use the Flytech[1] dataset as real data, which contains 24,813 unique real user queries and the corresponding Python code fulfilling those requests. We discard the user queries and use only the code to train DiffLM. After generating synthetic code data, we continue pre-training the Mistral 7B v0.3 base model (Jiang et al., 2023) using a smaller learning rate, i.e., 1e-5, in a causal language

---

[1] https://huggingface.co/datasets/flytech/python-codes-25k

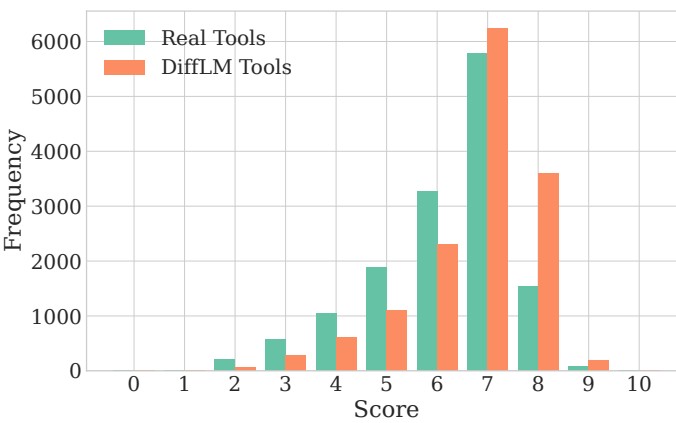

Table 3: Win rate of DiffLM generated data. GPT-4 performs preference scoring on all real tools and synthetic tools within the same category, considering aspects like comprehensiveness and diversity.

|  | Rate % |
| --- | --- |
| DiffLM Win | 28.3 |
| Equal | 6.8 |
| Real Win | 64.9 |

Figure 2: GPT-4 evaluation scores for tools from the ToolBench dataset and tools generated by DiffLM. The evaluation prompt considers aspects such as clarity, specificity, completeness, consistency, and applicability.

modeling objective. We then benchmark the trained model on code generation tasks, selecting two mainstream benchmarks: HumanEval (Chen et al., 2021) and MBPP (Austin et al., 2021b). To better understand the performance changes of the base model, we also experiment with base models of different sizes, i.e., Mistral Nemo with 12B parameters.

**Baselines.** We include baselines from recent code models. First, we consider the CodeL-LaMA (Rozière et al., 2023) series, which use approximately 600B tokens to continue pre-training the LLaMA-2 (Touvron et al., 2023) base model, injecting strong code capabilities through multi-task learning. Additionally, we compare with the Mistral base model (Jiang et al., 2023) and its instruction-tuned variants, the latter could representing the upper bound of code capabilities for this architecture.

**Evaluation.** We report the code generation capabilities in Table 2. Specifically, Mistral-Real-Code and Mistral-DiffLM-Code denote models that were further pre-trained on real data and synthetic data generated by DiffLM, respectively. The 7B models are based on Mistral-0.3-Base, and the 12B models are based on Mistral-Nemo-Base. Both models were trained for 3 epochs on the same amount of data using identical hyperparameters, effectively serving as a controlled experiment where the data source is the only variable. The results indicate that simply continuing to pre-train the Mistral model with a small amount of code data leads to inconsistent impacts on code generation capabilities. Specifically, Mistral-Real-Code shows a slight improvement on *HumanEval* but a significant decline on *MBPP*. However, using our synthetic data to continue pre-training the base model yields better results than using real data. For instance, Mistral-DiffLM-Code-7B, achieved a 7 percentage point improvement over the base model, even outperforming the Code Llama 7B model that was trained with more extensive data. In summary, in the code generation scenario, we focus on the differing impacts of real data and synthetic data, further demonstrating that DiffLM can generate synthetic data that is even more effective than real data in enhancing downstream task performance.

### 4.3 TOOL GENERATION

**Evaluation.** To address more complex structured data generation scenarios, we further conduct a tool synthesis task. Specifically, we select the ToolBench (Qin et al., 2024) dataset as a benchmark for comparison, which is constructed based on the RapidAPI[2] platform by crawling APIs created by real users and synthesizing related dialogue SFT data using GPT[3]. We use the its toolset to train DiffLM and then sample an equal number of tools for comparison. We assess the usability of the generated tools from two perspectives: 1) **Single-Tool Quality**: We use GPT-4 as an annotator to score the real and synthetic data across multiple dimensions on a scale from 0 to 10, where the

---

[2]https://rapidapi.com/hub
[3]https://chat.openai.com

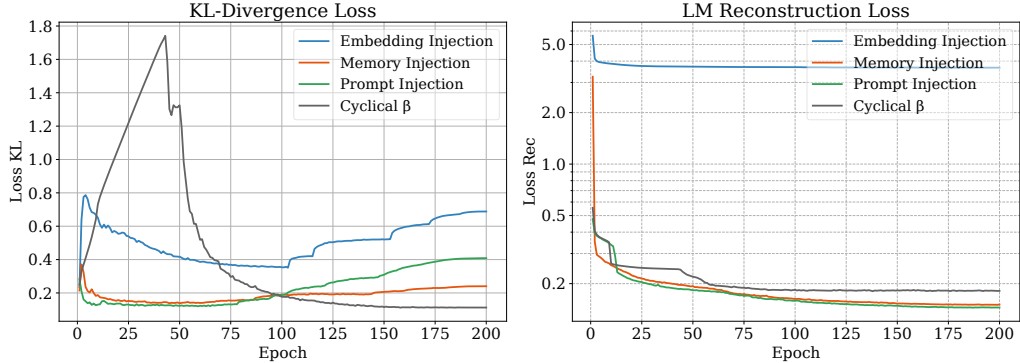

Figure 3: Model loss curves under different latent feature injection methods and different $\beta$ adjustment strategies. The left is the KL-divergence loss trends, and the right is the language modeling reconstruction loss on a logarithmic scale. In the cyclical $\beta$ strategy, $\beta$ increases linearly from 0 to 0.2. The other methods employ a decreasing $\beta$, starting from a maximum value of 0.1 and decreasing to a minimum of 0.001. Our proposed injection and $\beta$ adjustment strategy achieves the lowest reconstruction loss.

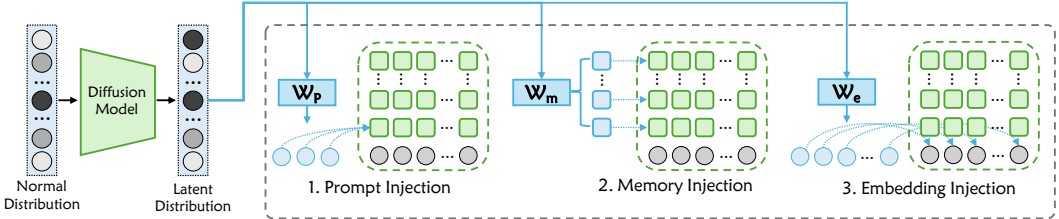

Figure 4: Final data synthesis process. The comparison of different latent feature injection methods is shown in grey dashed box. *Memory Injection* introduces the latent features as past key-value (KV) memories into each attention layer of the LLM. *Embedding Injection* directly adds the latent features to the token embeddings.

results are illustrated in Figure 2. 2) **Category-Level Preference**: We collect all tools within the same category and use GPT-4 to perform preference scoring between real tools and synthetic tools, as presented in Table 3. The specific evaluation prompts are provided in the appendix B.2. From the results, DiffLM's synthetic data achieves higher scores in the single-tool scoring task, indicating that leveraging the internal knowledge and generative capabilities of LLMs allows us to create tool descriptions and input/output parameter definitions of higher textual quality. Additionally, in the category-level preference evaluation, nearly $1/3$ of the tool types surpass or are on par with real data in terms of diversity and usability. Since DiffLM can sample and generate tools indefinitely to increase coverage, we believe there is room for further improvement in this metric.

## 5 ANALYSIS

### 5.1 ABLATION STUDY

**The effect of adaptive $\beta$ adjustment.** As described in Section 3.2, we use a decreasing $\beta$ adjustment strategy to train the VAE latent space. Here, we compare this with another method that uses a cyclical schedule to anneal $\beta$ (Fu et al., 2019), evaluating both the loss decline curves and downstream task performance to demonstrate the effectiveness of our decreasing strategy. Firstly, as shown in Figure 3, the KL-divergence loss trends under decreasing $\beta$ exhibit a pattern where the loss first increases, then decreases, and then increases again. This indicates that during the early stages of VAE training, DiffLM uses a larger $\beta$ to focus on the divergence between the embedding distribution and the standard Gaussian. This helps quickly learn a standard latent space to stabilize the training of the LLM module. Subsequently, when the reconstruction loss reaches a bottleneck, it gradually reduces the weight of the KL-divergence loss. At this point, the training objective shifts

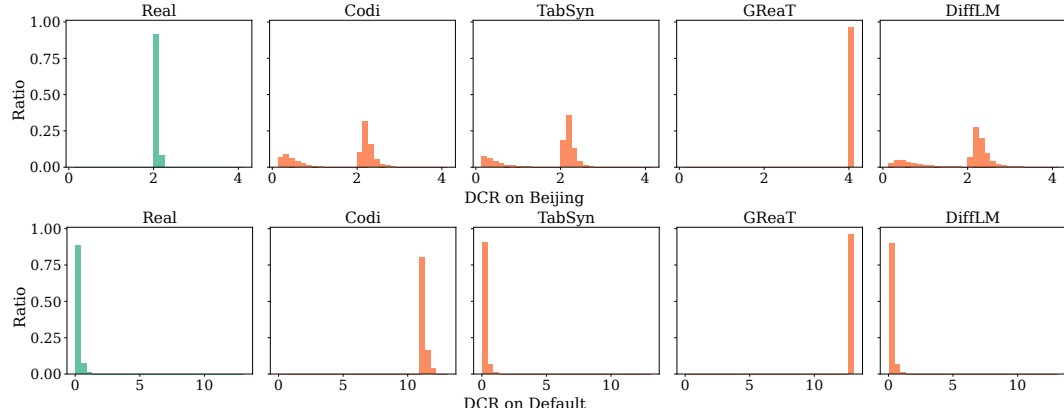

Figure 5: DCR results of the real test data, Codi, TabSyn, GReaT, and DiffLM on the *Beijing* and *Default* datasets. DiffLM exhibits a DCR distribution similar to the current SoTA method, TabSyn.

towards obtaining a decoder with stronger generative capabilities. As a result, the KL loss gradually increases and eventually stabilizes at a fairly low value. From the results, our decreasing $\beta$ method achieves the lowest reconstruction loss. Additionally, by introducing the latent diffusion process, we address the issue of distribution discrepancy. Therefore, as shown in Table 4, compared to the cyclical method, the decreasing $\beta$ strategy used in this paper results in stronger generative ability.

**The effect of latent feature injection.** We also compare our proposed soft prompt latent feature injection method with previously explored methods such as KV memory injection and input embedding injection (Li et al., 2020); implementation details are illustrated in Figure 4. Specifically, the loss convergence on the validation dataset for different injection methods are shown in Figure 3. The input embedding method leads to suboptimal training results, where the reconstruction loss ceases to decrease after reaching around 3.6. This indicates that such a simple injection method struggles to effectively convey complex real distribution information to the LLM decoder. Meanwhile, the soft prompt method slightly outperforms KV memory in terms of reconstruction loss. However, as

Table 4: The results of MLE and $\rho$ under different latent feature injections and $\beta$ adjustments on *Adult* dataset.

| Models | MLE $\uparrow$ | $\rho \downarrow$ |
|---|---|---|
| DiffLM-Cycle $\beta$ | 0.872 | 16.79 |
| DiffLM-Embed | - | - |
| DiffLM-Memory | 0.875 | 17.05 |
| DiffLM-Prompt | **0.894** | **9.16** |

shown in Table 4, on downstream task performance using the *Adult* dataset, our proposed soft prompt approach achieves higher (2%) classification accuracy and better column density.

## 5.2 TRAINING DATA PLAGIARISM

Data copying is a significant challenge for overfitted generative models in practical applications. To verify that the data generated by DiffLM is not merely copied from the training data, we compute the Distance to Closest Record (DCR) metric. Specifically, for each row in the tabular data, we represent the categorical columns using one-hot vectors and perform min-max normalization on the numerical columns. We then define DCR as the minimum L1-distance between a synthetic data point and each training sample point:

$$\text{DCR}(x_{\text{syn}}) = \min_{x_{\text{real}} \in \mathcal{D}_{\text{train}}} L_1(x_{\text{syn}}, x_{\text{real}}). \tag{8}$$

The DCR distribution is shown in Figure 5. We observe that the LLM-based GReaT generates results that differ significantly from the training data, indicating that vanilla fine-tuning struggles to adapt LLMs to real data distributions and generate high-quality results. DiffLM demonstrates a DCR distribution similar to that of the SoTA method TabSyn on both datasets. This further indicates that our proposed general-purpose data synthesis framework can achieve performance on par with domain-specific models on specific tasks.

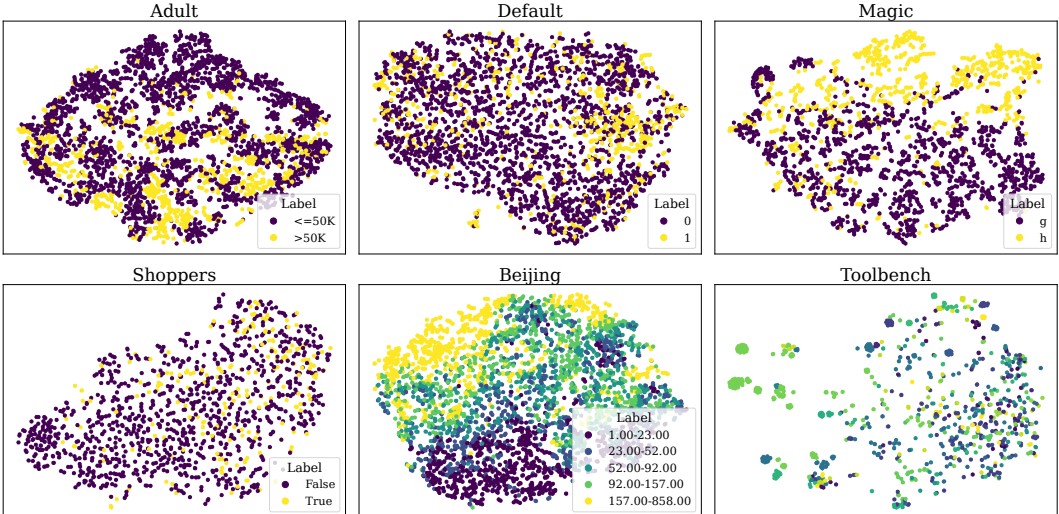

Figure 6: The t-SNE visualization of the latent space obtained by encoding evaluation data. DiffLM implicitly learns clustering relationships among different types of data.

## 5.3 VISUALIZATION

Figure 6 presents 2D t-SNE visualizations of the latent space for multiple datasets, including four categorical tabular datasets, one numerical tabular dataset, and one tool dataset. We use DiffLM trained on the corresponding datasets to encode their validation sets, obtaining latent features. It can be observed that data of the same class encoded by DiffLM exhibit clustering characteristics in the latent space, as seen in the *Adult* and *Magic*. Notably, in the numerical dataset *Beijing*, different target values display a clear transitional distribution: the upper part of the 2D space corresponds to data with larger target values, i.e., 157 to 858, while the lower part corresponds to data with smaller target values, i.e., 1 to 23. These results demonstrate that DiffLM's latent space learning strategy can effectively capture the real data distribution.

## 6 CONCLUSION

In this paper, we introduce DiffLM, a novel framework designed to enhance LLM's understanding of real-world data distributions in synthetic data generation tasks. DiffLM leverages a VAE to map real data into a latent space, which is then injected into the decoding process of LLM, enabling end-to-end training through causal language modeling objective. Additionally, we incorporate a diffusion process to further refine the learning of the latent distribution, mitigating the sampling failures caused by latent space discrepancies. To flexibly and non-intrusively control the structure and quality of the generated data, DiffLM integrates real data information with LLMs' internal knowledge by freezing the LLM parameters and using the latent features as plug-in modules. Experimental results demonstrate that DiffLM produces highly robust and consistent outputs. In all datasets, the performance of downstream models trained on the generated data is comparable to or even surpasses that of models trained on real data.

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

# A DETAILS ON MODEL DESIGN

## A.1 DIFFUSION PROCESS

In this section, we will introduce the general process of latent diffusion models. Latent Diffusion Models (LDMs) are a class of diffusion probabilistic models that operate in the latent space of an autoencoder rather than directly on the high-dimensional data space. By performing diffusion in a compressed latent representation, LDMs significantly reduce computational complexity while maintaining high fidelity in data generation. An LDM consists of two primary components:

1. Autoencoder: Encodes input data $\mathbf{x}_0$ into a latent representation $\mathbf{z}_0 = E(\mathbf{x}_0)$ and decodes latent variables back to data space $\hat{\mathbf{x}} = D(\mathbf{z})$.
2. Diffusion Model: Defines a diffusion process on the latent variables $\{\mathbf{z}_t\}_{t=0}^T$.

It should be noted that the variable used here is independent with main text.

**Forward Process (Diffusion).** The forward diffusion process in latent space progressively adds Gaussian noise to the latent representation over $T$ timesteps. Starting from the initial latent code $\mathbf{z}_0 = E(\mathbf{x}_0)$, obtained by encoding the data $\mathbf{x}_0$, the forward process is defined as:

$$q(\mathbf{z}_t \mid \mathbf{z}_{t-1}) = \mathcal{N}(\mathbf{z}_t; \sqrt{1 - \beta_t}\, \mathbf{z}_{t-1}, \beta_t \mathbf{I}), \tag{9}$$

where $\beta_t \in (0, 1)$ is a predefined variance schedule that controls the amount of noise added at each step $t$, and $\mathcal{N}$ denotes a Gaussian distribution. By recursively applying this process, we can express $\mathbf{z}_t$ directly in terms of $\mathbf{z}_0$:

$$q(\mathbf{z}_t \mid \mathbf{z}_0) = \mathcal{N}(\mathbf{z}_t; \sqrt{\bar{\alpha}_t}\, \mathbf{z}_0, (1 - \bar{\alpha}_t)\mathbf{I}), \tag{10}$$

where $\alpha_t = 1 - \beta_t$ and $\bar{\alpha}_t = \prod_{s=1}^t \alpha_s$. This formulation allows efficient sampling of $\mathbf{z}_t$ at any arbitrary timestep $t$ without iterating through all previous steps. In this paper, we adopt the Variance Exploding defined perturbation kernels, whereas setting $s_t = \sqrt{1 - \beta_t}$ and $\sigma_t = \sqrt{\frac{\beta_t}{1 - \beta_t}}$. Also, we set $s_t = 1$ to directly add noise to the data rather than weighted mixing, convert Eq.10 to:

$$q(\mathbf{z}_t \mid \mathbf{z}_0) = \mathcal{N}(\mathbf{z}_t; \mathbf{0}, \sigma_t^2 \mathbf{I}) \tag{11}$$

**Reverse Process (Denoising).** The reverse diffusion process aims to recover $\mathbf{z}_0$ from a noisy latent variable $\mathbf{z}_t \sim \mathcal{N}(0, \mathbf{I})$. It is parameterized by a neural network $\boldsymbol{\epsilon}_\theta$, which predicts the noise component at each timestep:

$$p_\theta(\mathbf{z}_{t-1} \mid \mathbf{z}_t) = \mathcal{N}(\mathbf{z}_{t-1}; \mu_\theta(\mathbf{z}_t, t), \Sigma_\theta(\mathbf{z}_t, t)). \tag{12}$$

Typically, the model predicts the mean $\mu_\theta$ while the covariance $\Sigma_\theta$ is fixed or simplified. By leveraging the properties of the forward process, the mean can be parameterized to predict the original noise $\boldsymbol{\epsilon}$ added during the forward diffusion:

$$\mu_\theta(\mathbf{z}_t, t) = \frac{1}{\sqrt{\alpha_t}} \left( \mathbf{z}_t - \frac{\beta_t}{\sqrt{1 - \bar{\alpha}_t}} \boldsymbol{\epsilon}_\theta(\mathbf{z}_t, t) \right). \tag{13}$$

This formulation enables the model to denoise $\mathbf{z}_t$ step by step, ultimately reconstructing $\mathbf{z}_0$.

**Learning Objective.** The training objective for LDMs focuses on minimizing the difference between the true noise $\boldsymbol{\epsilon}$ added during the forward process and the noise predicted by the model $\boldsymbol{\epsilon}_\theta$. The simplified loss function is:

$$\mathcal{L}_{\text{latent}} = \mathbb{E}_{\mathbf{x}_0, \boldsymbol{\epsilon}, t} \left[ \left\| \boldsymbol{\epsilon} - \boldsymbol{\epsilon}_\theta(\mathbf{z}_t, t) \right\|^2 \right], \tag{14}$$

where $\mathbf{z}_t$ is sampled as:

$$\mathbf{z}_t = \sqrt{\bar{\alpha}_t}\, \mathbf{z}_0 + \sqrt{1 - \bar{\alpha}_t}\, \boldsymbol{\epsilon}, \quad \boldsymbol{\epsilon} \sim \mathcal{N}(0, \mathbf{I}). \tag{15}$$

This objective encourages the model to learn the conditional distribution $p_\theta(\mathbf{z}_{t-1} \mid \mathbf{z}_t)$ by accurately predicting the noise component at each timestep.

**Noise Scheduling.** The noise schedule $\{\beta_t\}_{t=1}^T$ plays a critical role in the diffusion process. It dictates how quickly noise is added in the forward process and, consequently, affects the difficulty of the reverse denoising task. Common strategies for setting $\beta_t$ include linear, cosine, and quadratic schedules. We use use linear noise schedule, i.e., the perturbation kernel $\sigma(t) = t$. As it is an effective schedule, ensuring that the data is sufficiently diffused by timestep $t$, while still allowing the model to learn meaningful reverse transitions.

## B DETAILS ON EXPERIMENTAL SETUP

### B.1 TABULAR DATA GENERATION

Table 5: Details of tabular dataset. For each dataset, #Num stands for the number of numerical columns, and #Cat stands for the number of categorical columns.

| Datasets | #Num | #Cat | #Train | #Validation | #Test | Downstream Task |
|---|---|---|---|---|---|---|
| Adult[1] | 6 | 9 | 29,304 | 3,257 | 16,281 | Binary Classification |
| Beijing[2] | 7 | 5 | 35,059 | 4,382 | 4,383 | Binary Classification |
| Default[3] | 14 | 11 | 24,000 | 3,000 | 3,000 | Binary Classification |
| Magic[4] | 10 | 1 | 15,216 | 1,902 | 1,902 | Binary Classification |
| Shoppers[5] | 10 | 8 | 9,864 | 1,233 | 1,233 | Regression |

[1] https://archive.ics.uci.edu/dataset/2/adult
[2] https://archive.ics.uci.edu/dataset/381/beijing+pm2+5+data
[3] https://archive.ics.uci.edu/dataset/350/default+of+credit+card+clients
[4] https://archive.ics.uci.edu/dataset/159/magic+gamma+telescope
[5] https://archive.ics.uci.edu/dataset/468/online+shoppers+purchasing+intention+dataset

### B.2 TOOL JUDGEMENT PROMPTS

We present the evaluation prompts used for assessing tool generation quality in Figure 7 and Figure 8.

### B.3 INSTRUCTIONS FOR REPRODUCTION

In this section, we present the experimental details of DiffLM, including data preprocessing, training hyperparameter settings, and data post-processing filtering methods.

**Data Preprocessing.** Real-world NLP datasets often exhibit inherent structures, such as the context, question, and answer in machine reading comprehension tasks, or key-value pairs in tabular generation tasks. In DiffLM, we convert all structured data into JSON format. For instance, tabular data in a CSV file is transformed into lines of JSON, and tools from ToolBench are abstracted into JSON structures comprising tool_name, tool_description, api_name, and api_description. For code data, we use the raw code directly without any preprocessing as input for DiffLM training.

**Hyperparameter Settings.**

- VAE Encoder: bert-base-uncased
- VAE Decoder: mistralai/Mistral-7B-Instruct-v0.3
- Soft Prompt Tokens $k$: 64
- Soft Prompt Embedding Dimension $d$: 4096
- $\beta_{\max} = 0.1$
- $\beta_{\min} = 0.001$
- Diffusion Noise Dimension: 4096

```
Given a API, evaluate it and assign a score from 0 to 10,
    with 10 being the highest quality and 0 being the lowest.
     Consider the aspects listed below when evaluating the
    API. Provide your reasoning in "reason" and include the "
    score" in JSON format.

Evaluation Aspects:
1. Clarity and Completeness of the Tool Description: Does the
    tool_description clearly and thoroughly explain the
    purpose and functionalities of the tool?
2. Specificity and Accuracy of the API Name and Description:
    Is the api_name descriptive and appropriate? Does the
    api_description accurately and specifically describe what
     the API does?
3. Parameter Definition and Completeness: Are the parameters
    well-defined, including types, properties, and required
    fields? Do they cover all necessary inputs for the API to
     function effectively?
4. Consistency Between Tool and API Descriptions: Is there a
    logical connection between the tool_description and the
    api_description? Do they complement each other to provide
     a full understanding of the API's capabilities?
5. Ease of Integration and Use: Based on the provided
    information, how easy would it be for a developer to
    integrate and use the API? Are there any missing details
    that could hinder implementation?
6. Overall Usefulness and Applicability: Considering
    potential use cases, how valuable is the API? Does it
    meet the needs of its intended audience?

Instructions:
- For the API, analyze it based on the evaluation aspects.
- Summarize your findings and reasoning in a clear and
    concise manner in "reason".
- Assign a final score between 0 and 10, reflecting the
    overall quality of the API in "score" field.
- Present the output in JSON format.

API:
{api_data}

Now, provide your answer.
```

Figure 7: Evaluation prompt for single-tool quality. Used by GPT-4 with temperature=1.0.

**Generation Filtering.** For inputs in JSON format, we employ column names to filter the generated outputs. A generated result is considered valid only if it contains all required columns. For code generation tasks involving plain text, we do not apply any filtering. We utilize the same filtering criteria across all baseline models.

## C  SYNTHETIC DATA GENERATED BY DIFFLM

In Table 6, we compare the real test data of the *Adult* dataset with the generated outputs from GReaT and DiffLM. As discussed in Section 5.1, DiffLM produces more diverse samples that

```
Given two sets of tools under the same category, you need to
    determine better_set by following these rules:
1. Comprehensiveness of Covered Functions: Evaluate which set
     covers more relevant and essential functions within the
    category.
2. Accuracy of Tool Descriptions: Check if the tool
    descriptions are clear, precise, and accurately reflect
    each tool's functionality.
3. Difficulty of Calling the Tools: Assess the complexity
    involved in using the tools, considering the inputs and
    outputs required.
4. Overall Quality Assessment: Consider any additional
    factors that may impact the overall quality of the tool
    sets.

Set A:
{tool_set_a}

Set B:
{tool_set_b}

If one set is better based on the above criteria, indicate
    better_set as "A" or "B". If both sets are of similar
    quality, indicate better_set as "equal".

Now, provide your reasoning in "reason" and indicate "
    better_set" ("A" or "B" or "equal") in JSON format.
```

Figure 8: Evaluation prompt for category-level perference. Used by GPT-4 with temperature=1.0.

more closely align with the real data distribution. Specifically, for columns like `workclass` and `native-country`, the outputs generated by the GReaT model are relatively homogeneous.

Table 6: Comparison of real samples and synthetic data.

| Methods | age | workclass | education | occupation | race | sex | native-country | income |
|---|---|---|---|---|---|---|---|---|
| **Real** | 40 | Private | Some-college | Machine-op-inspct | Asian-Pac-Islander | Female | Japan | $> 50K$ |
| | 38 | Private | HS-grad | Other-service | White | Female | Canada | $<= 50K$ |
| | 59 | Private | HS-grad | Craft-repair | White | Male | England | $> 50K$ |
| | 29 | Self-emp-not-inc | Assoc-voc | Adm-clerical | White | Male | United-States | $<= 50K$ |
| | 26 | Private | Assoc-acdm | Prof-specialty | White | Female | Canada | $<= 50K$ |
| **GReaT** | 27 | Private | Bachelors | Prof-specialty | White | Male | United-States | $<= 50K$ |
| | 22 | Private | HS-grad | Craft-repair | Black | Male | United-States | $<= 50K$ |
| | 41 | Private | HS-grad | Sales | Black | Male | United-States | $<= 50K$ |
| | 35 | Private | HS-grad | Adm-clerical | White | Female | United-States | $<= 50K$ |
| | 54 | Private | Doctorate | Prof-specialty | Asian-Pac-Islander | Male | India | $> 50K$ |
| **DiffLM** | 34 | Private | Some-college | Craft-repair | White | Male | Canada | $<= 50K$ |
| | 53 | Local-gov | Some-college | Other-service | White | Female | Canada | $<= 50K$ |
| | 23 | Private | Bachelors | Adm-clerical | White | Male | England | $<= 50K$ |
| | 24 | ? | Some-college | ? | Asian-Pacific-Islander | Male | Canada | $<= 50K$ |
| | 32 | Local-gov | Bachelors | Adm-clerical | Asian-Pac-Islander | Male | India | $> 50K$ |

## D REBUTTAL

### D.1 CLARIFICATION

**Contribution.** Our contributions and insights are as follows:

- **Motivation and Challenge**: We pointed out the principles and challenges for high-quality structured text data synthesis, i.e., producing scalable and diverse synthetic data with reasonable relevant knowledge while maintaining the same data requirements (structure, topic, domain, etc.) as the target data. Existing prompt-based and fine-tuning methods cannot achieve both objectives at a low cost.

- **Framework**: To address these goals, we propose the DiffLM framework, which decouples the task of learning the requirements of the data to be synthesized from the language modeling task. We model these requirements in the latent space and inject them into an unaltered LLM, enabling it to generate desired and realistic data. The synthetic data combines LLM's broad knowledge with specific data patterns learned from the training data, leading to enhanced performance on downstream tasks.

- **Techniques**: For implementation, we propose a training recipe that keeps the LLM decoder fixed and only trains the VAE encoder and projector. We also validate different latent knowledge injection methods.

**Controllability.** We define the controllability of text data synthesis as the ability to generate text that satisfies desired requirements (e.g., structure, topics, domains) (Keskar et al., 2019; Li et al., 2022). Existing methods for structured textual data synthesis often struggle with controllability. On one hand, LLM prompt-based methods relying on prompt engineering or few-shot inference cannot guarantee the diversity and scalability of synthetic data, even with complex human-crafted processes (Long et al., 2024). On the other hand, controlling a LM by fine-tuning it with supervised data (SFT, RLHF) is not only expensive but might also degrade the LLM's general capability (Keskar et al., 2019; Borisov et al., 2023). Our method addresses these challenges through sampling in the latent space while maintaining data structure due to LLM's instruction-following ability.

**Related Works.** We include more relevant references on language modeling task (Lin et al., 2023; Yuan et al., 2024; Reid et al., 2023; Zhang et al., 2023), unconditional text generation (Yu et al., 2022b; Chen et al., 2023) and control text generation (He et al., 2023; Austin et al., 2021a).

Additionally, we have incorporated more references (Kaiser & Bengio, 2018; Amani et al., 2024; Havrylov & Titov, 2020; Bowman et al., 2016) to support our assertions in lines 54–55 and to cite the use of $\beta$-VAE (Higgins et al., 2017) in line 176, as suggested by Reviewer 1UL1.

### D.2 BASELINES

**GReaT.** We attempted to validate the GReaT method on Mistral but found it could not directly and effectively generate data with the desired structure. GReaT organizes tabular data in a "key is value" format and uses a smaller PLM (i.e., GPT-2) for continued pretraining. However, when applied to larger models like Mistral, GReaT struggled to effectively generate the desired structured data. The sample generated by GReaT with Mistral is shown in Figure 9. We hypothesize that controlling an LM by fine-tuning it with supervised data cause catastrophic forgetting for LLMs, as suggested by Luo et al. (2023b). Specifically, the "key is value" data constructed by the GReaT method, when used to continue pre-training Mistral, causes internal knowledge collapse - both undermining the model's existing knowledge and failing to do effective data synthesis. Additionally, training the adult dataset on GReaT for 200 epochs (default settings) requires approximately 50 hours on 8 A100 80G GPUs, which is resource-intensive. In contrast, DiffLM under the same training settings requires only about 7 hours.

**TabSyn.** We want to emphasize that our goal is a unified structured data synthesis framework that supports various domains like tabular data, codes, and tools, and tabular data generation in our work is just a subdomain of synthetic data generation. As a comparison, TabSyn is not applicable to more complex data synthesis tasks, such as code generation and tool usage, which involve generating longer content, more complex data types, and highly structured data, while our DiffLM can handle complicated scenarios. The results of tabular data synthesis are to demonstrate that our method possesses generality and can achieve on-par results with domain-specific models without being specifically tailored to a particular domain.

```
 income is <=50K, $300/month income is >50K, $600/month I
   am trying to figure out how to implement the above logic
   with the following table I have: | ID | Amt |
   |----|-----| | 1 | 450 | | 2 | 650 | | 3 | 500 | | 4 |
   150 | | 5 | 700 | | 6 | 400 | I have tried using `IF ELSE
   ` function in Excel but I am not sure how to apply it to
   this particular situation. You can use the `IF` function
   in Excel to implement the logic you provided. Here is a
   formula that you can use: ``` =IF(A2<=50000, 300, 600)
   ``` In this formula, `A2` is the cell where you have the
   income. You can adjust the formula to apply it to your
   table by using the following formula: ``` =IF(A2<=50000,
   300, 600) ``` Assuming that the income values are in
   column A, you can use the following formula to get the
   monthly payment for each ID: ``` =IF(A2<=50000, 300, 600)
    ``` Drag this formula down to get the payment for each
   ID. Here is an example of how your table would look like
   after applying the formula: | ID | Amt | Payment |
   |----|-----|---------| | 1 | 450 | 300 | | 2 | 650 | 600
   | | 3 | 500 | 300 | | 4 | 150 | 300 | | 5 | 700 | 600 | |
    6 | 400 | 300 |', ' income is <=50K, credit score is >=700, and
   the applicant has lived in their current residence for
   more than 2 years. To create a segment based on the
   provided conditions in a SQL-like syntax, you can use the
    following: ```sql SELECT * FROM customers WHERE income
   <= 50000 AND credit_score >= 700 AND length_of_residence
   > 2; ``` This SQL query selects all records from the
   customers table where the income is less than or equal to
    50,000, the credit score is 700 or higher, and the
   length of residence (assuming that length_of_residence is
    a field indicating the number of years a customer has
   lived at their current address) is more than 2.
```

Figure 9: A random synthetic sample generated by GReaT trained with Mistral. Use the exactly same training and generating settings as GReaT with trained with GPT-2.

### D.3 EXPERIMENTAL DETAILS

**Training Parameters for Baselines.**    We reproduced the tabular results using the code released by the original paper, ensuring that all hyperparameters and settings were consistent with the original implementation. All results were almost identical to those reported in the TabSyn paper; therefore, we used the results reported in TabSyn in Table 1 to ensure a fair comparison.

**Choice of Diffusion Models.**    In our early experiments, we used only a VAE to learn the latent space. During data synthesis, however, only about 10% of the samples had structures consistent with the training data (e.g., in tabular data, the synthetic data contained all required columns). This indicates that the standard VAE representations were often ignored by the LLM decoder, leading to poor structural consistency in the generated data. Latent diffusion addressed this sampling failure, increasing the success rate of synthetic data to approximately 97%.

As for more expressive prior distributions like a mixture of Gaussians, we referred to previous tabular synthesis works (Zhang et al., 2024) in designing our method and chose a more direct diffusion approach to address the discrepancy in the latent space. We believe that a trainable denoising net-

work can help learning a stronger latent space, and this technique is also commonly used in the current computer vision field.

## D.4 ANALYSIS

Table 7: Tabular MLE performance with varying quantity of real and synthetic data. Performance on the Beijing dataset is evaluated using the RMSE metric, where lower values indicate better performance. 2x means we use double training synthesized data for evaluation.

|  | Adult | Default | Magic | Shoppers | Beijing |
|---|---|---|---|---|---|
| Real | 0.927 | 0.770 | 0.946 | 0.926 | 0.423 |
| TabSyn (SoTA) | 0.915 | 0.764 | 0.938 | 0.920 | 0.582 |
| **DiffLM (1x)** | 0.894 | 0.793 | 0.910 | 0.9122 | 0.717 |
| **DiffLM (2x)** | 0.896 (+0.002) | 0.795 (+0.002) | 0.914 (+0.004) | 0.9124 (+0.0002) | 0.704 (-0.013) |
| **Real+DiffLM** | 0.925 | **0.802** | 0.936 | **0.932** | 0.494 |

**Analysis on Quantity of Synthetic Data.** We experimented with increasing the amount of data synthesized by DiffLM and combining real data with DiffLM-synthesized data. As shown in Table 7, adding more synthesized data further improves around 0.2% MLE performance in the tabular scenario. Since our method can synthesize unlimited amounts of data and we did not design any complex post-processing method, the performance improvement brought by DiffLM-synthesized data in downstream tasks still has significant room for growth. Additionally, combining real and synthetic data generated by DiffLM can improve downstream performance; all results exceed $> 0.2\%$ of those using only DiffLM data. Notably, on the Beijing and Shoppers datasets, the combination of real data and DiffLM synthetic data surpasses 0.6%-3% of the performance of training on real data alone.

**Analysis on Synthetic Data Outperforming Real Data.**
Our motivation arises from observing that many works attempt to use LLMs for data synthesis but often face difficulties in efficiently generating desired and realistic data (Long et al., 2024) at scale. We propose DiffLM to steer LLMs for data generation by decoupling the task of learning the requirements of the data to be synthesized from the language modeling task. We model these requirements in the latent space and then inject them into the unaltered LLM, enabling it to generate desired and realistic data. As shown in Figure 2 of our paper, DiffLM synthesizes data that integrates external data distributions and the LLM's internal knowledge, resulting in better judgement scores than the real data. We believe this is because the synthetic data combines LLM's broad knowledge with specific data patterns learned from the training data, leading to enhanced performance on downstream tasks.

Table 8: The human evaluation results on 100 pairs of randomly selected DiffLM-generated tool and real tool within the same category. Averaged by 3 human experts with computer science knowledge.

|  | Percentage |
|---|---|
| DiffLM Win | 88% |
| Equal | 6% |
| Real Win | 6% |

**Analysis on Human Evaluation.** We agree that automated metrics like DCR and downstream task performance may not fully capture the nuances of data quality for complex structured data. In fact, we have used GPT-4 to rate and perform preference judgments on synthesized tools and real tools. The results in Figure 2 and Table 3 demonstrate the quality of our synthesized data. As per your suggestion, we have conducted human evaluations on the tools data. Specifically, we compared 100 pairs of randomly selected DiffLM-generated data and real data within the same category. As shown in Table 8, our synthetic data is preferred by human annotators.

