# OpenReview forum: "DiffLM: Controllable Synthetic Data Generation via Diffusion Language Models"
_ICLR.cc/2025/Conference — Submitted to ICLR 2025_

### Official Review · Reviewer_1UL1 · 2024-11-04

**Soundness:** 2
**Presentation:** 2
**Contribution:** 2
**Rating:** 5
**Confidence:** 3

**Summary:**

This paper investigates methods for generating synthetic data samples from a pre-trained large language model (LLM) using diffusion-based methods to approximate the underlying data distribution. The authors explore the limitations of standard fine-tuning approaches in capturing true data distribution and propose an alternative involving a VAE (Variational Autoencoder) setup where a latent representation encodes the distribution and a conditioned LLM decoder generates samples.

The approach can be summarized as follows:

1. Fine-tuning an LLM to generate samples often results in samples diverging from the original data distribution (cited in lines 90-96).
2. To address this, the authors propose using a VAE with a learned latent representation to capture the data distribution, conditioning the LLM decoder on these representations through a soft prompt generated by an MLP.
3. Standard VAE setups can face issues with strong decoders, where the encoder vector may be ignored. To mitigate this, the authors introduce denoising diffusion steps in the training and employ a $\beta$-VAE for additional control over the KL-divergence term via the adjustable $\beta$ parameter.

Experiments span three task types across seven datasets.

**Strengths:**

The paper is structured clearly, addressing an important problem in synthetic data generation.

Experimental results indicate performance improvements, and the visualizations are well-crafted, aiding comprehension of the methods and outcomes.

Ablation studies on injection methods provide useful insights

**Weaknesses:**

The work lacks a comprehensive ablation and stronger baseline comparisons. Although the authors conduct ablations, they focus on minor details like the $\beta$ adjustments and design choice of latent feature injection, rather than the main components of the proposed pipeline. Key design choices lack justification and comparative analysis, for instance the choice of doing denoising diffusion as I describe later. Despite references to the limitations of standard fine-tuning with sampling, the paper does not provide a thorough comparison of these methods against the proposed framework, showing enough evidence on what precise problem of finetuning and sampling is resolved (please check the questions for more on this.)

There is also a lack of evidence for key claims. In Lines 54-55, the paper mentions that standard VAE representations tend to be ignored by the LLM, driving the need for the diffusion-based approach. However, there is insufficient evidence or citation support for this assertion. Similar issues have been addressed in prior works (e.g., [2, 4, 5]).

Similarly in lines 195-196, if **the core issue** is a discrepancy between the encoder’s posterior and prior, citations or empirical evidence are needed to support this claim. Additionally, simpler solutions, such as adopting a more expressive prior distribution (e.g., Mixture of Gaussians), could have been discussed as alternatives to diffusion. I think the choice of diffusion models is not justified in the current write up.

**Some Minor Recommendations and Editorial Notes**

- I recommend citing the original $\beta$-VAE work since you're using this technique (reference [3]).
- **Conditional Generation with LLMs and VAEs**: Since the paper explores variational autoencoding with LLMs, you could consider to reference prior work on conditional generation with hard and soft tokens in LLMs with auto-encoders. Related work such as [1, 2, 4, 5] would provide readers with a broader context and highlight parallels to existing methods.

**Questions:**

1. Line 153 mentions that $D \cap D_{syn} = \emptyset$, yet it is unclear to me how Figure 5 demonstrates this condition is met for the proposed method but not for LLM fine-tuning.
2. For the fine-tuning baseline, the results may be highly dependent on training parameters (e.g., number of epochs, memorization, sampling strategy). What steps were taken to ensure a fair comparison?


**References**

1. Kaiser and Bengio. "Discrete Autoencoders for Sequence Models."
2. Amani et al. "Symbolic Autoencoding for Self-Supervised Sequence Learning."
3. Higgins et al. "$\beta$-VAE: Learning Basic Visual Concepts with a Constrained Variational Framework."
4. Havrylov and Titov. "Preventing Posterior Collapse with Levenshtein Variational Autoencoder."
5. Bowman et al. "Generating sentences from a continuous space".

---

> ### Author Response · Authors · 2024-11-24
> **Response to Reviewer 1UL1**
>
> Thank you for your constructive comments and the related work you provided. We address your concerns as follows:
>
> **Fine-tuning Baselines:**
>
> We use GReaT as the baseline because it is the current SoTA tabular synthesis method based on PLMs (GPT-2). However, fine-tuning larger LLMs (i.e., LM with more than 7B parameters) for structured data synthesis is not trivial. We exploited GReaT with stronger LLMs by replacing GPT-2 with Mistral, but it did not effectively generate data with the desired structure. The synthetic sample is shown in follow. **We hypothesize that controlling an LM by fine-tuning it with supervised data cause catastrophic forgetting for LLMs, as suggested by Luo et al[1]**. Specifically, the "key is value" data constructed by the GReaT method caused internal knowledge collapse when continuing to pre-train Mistral. In summary, this not only impaired the model's inherent knowledge but also failed to learn data synthesis effectively.
>
> Synthetic data generated by GReaT with Mistral:
>
> ```
> <s> income is <=50K, $300/month  income is >50K, $600/month  I am trying to figure out how to implement the above logic with the following table I have:  | ID | Amt | |----|-----| | 1  | 450 | | 2  | 650 | | 3  | 500 | | 4  | 150 | | 5  | 700 | | 6  | 400 |  I have tried using `IF ELSE` function in Excel but I am not sure how to apply it to this particular situation.......</s>
> ```
>
> Regarding the training parameters, we reproduced the tabular results using the code released by the original paper, ensuring that all hyperparameters and settings were consistent with the original implementation. All results were almost identical to those reported in the TabSyn paper; therefore, we used the results reported in TabSyn in Table 1 to ensure a fair comparison.
>
> **Justification for the Choice of Diffusion Models:**
>
> In our early experiments, we used only a VAE to learn the latent space. During data synthesis, however, only about 10% of the samples had structures consistent with the training data (e.g., in tabular data, the synthetic data contained all required columns). This indicates that the standard VAE representations were often ignored by the LLM decoder, leading to poor structural consistency in the generated data. Latent diffusion addressed this sampling failure, increasing the success rate of synthetic data to approximately 97%.
>
> As for more expressive prior distributions like a mixture of Gaussians, we referred to previous tabular synthesis works[2] in designing our method and chose a more direct diffusion approach to address the discrepancy in the latent space. We believe that a trainable denoising network can help learning a stronger latent space, and this technique is also commonly used in the current computer vision field.
>
> **Explanation of DCR, Figure 5:**
>
> DCR (Distance to Closest Record) is used to measure the closest distance between the target data and the training data. In Figure 5, we present the DCR histograms of the test data and the synthetic data. Intuitively, a better DCR distribution should be closer to that of the test data. From the figure, we can see that the synthetic data generated by DiffLM did not copy the training data (all DCRs are greater than 0), while maintaining a DCR distribution similar to that of the test data compared to the finetuning based method, i.e., GReaT.
>
> **Ablation experiments:**
>
> Thank you for pointing this out. We are currently working on ablations for the denoising network. Regarding the fine-tuning baselines, they performed poorly with LLMs like Mistral, as mentioned above. We will add the necessary descriptions and analyses to the paper to clarify these points.
>
> **Related Works:**
>
> Thank you for the references you provided. We will add them to the paper and discuss how they relate to our work.
>
> Reference:
>
> [1] Luo et al. An Empirical Study of Catastrophic Forgetting in Large Language Models During Continual Fine-tuning. arXiv 2023.
>
> [2] Zhang et al. Mixed-Type Tabular Data Synthesis with Score-based Diffusion in Latent Space. ICLR 2024.

---

> > ### Author Response · Authors · 2024-11-28
> > **Experimental results of ablation**
> >
> > Dear reviewer, to address your concern, we conducted ablation experiments on the denoising network. Specifically, to ensure a fair comparison with prior works (e.g., TabDDPM, TabSyn), DiffLM employed a simple MLP denoising network with the following setup: 1) sinusoidal timestep embeddings, 2) a 5-layer MLP denoising network, and 3) EDM training loss.
> >
> > In Table A, we present the results of our ablation experiments, which involved: 1) replacing the sinusoidal timestep embedding with a random-frequency-based timestep embedding (DiffLM-rnd), 2) increasing the number of MLP layers to 10 (DiffLM-10L-MLP), and 3) substituting the MLP with an attention-based network (DiffLM-Attn). The results demonstrate that DiffLM's performance can be further improved with more complex denoising networks. For instance, increasing the MLP layers from 5 to 10 improved the MLE performance on the Adult dataset from 0.894 to 0.904. Moreover, replacing the MLP with an attention-based network further boosted performance to 0.906.
> >
> > **Table A:** Tabular MLE performance on Adult datasets.
> > |                | Adult MLE |
> > | :------------- | :-------- |
> > | TabSyn(SoTA)   | 0.915    |
> > | DiffLM         | 0.894    |
> > | DiffLM-rnd | 0.896    |
> > | DiffLM-10L-MLP | 0.904    |
> > | DiffLM-Attn    | 0.906    |
> >
> > These results confirm that DiffLM is an effective framework capable of further enhancement. However, it should be noted that we kept the denoising network simple in paper to ensure a fair comparison with prior works. We will include the details of the denoising network and the ablation experiments, along with the corresponding results table, in the Appendix of the revised version of the paper.
> >
> > Considering the Rebuttal deadline is approaching, we sincerely hope to receive your response. If you have any questions or concerns regarding our explanation, please do not hesitate to contact us. Your response is of great significance to us for improving this work, and we look forward to hearing from you.

---

### Official Review · Reviewer_rfz8 · 2024-11-04

**Soundness:** 3
**Presentation:** 3
**Contribution:** 3
**Rating:** 6
**Confidence:** 4

**Summary:**

The paper introduces a novel framework, named DiffLM, for controllable synthetic data generation using diffusion models and LLMs. It addresses challenges in synthetic data generation by combining a variational autoencoder (VAE) with a latent diffusion module to enhance the understanding of target data distributions. The model was tested across different type of datasets, including tabular, code, and tool data. Results showed that DiffLM could surpass real data performance in some cases.

**Strengths:**

1. DiffLM effectively captures real-world data distributions through a combination of VAE and diffusion models, which enables the generation of high-quality synthetic data.
2. By using soft prompt injection, DiffLM allows for easy control over generated data characteristics without retraining the language model, which makes it adaptable across diverse data types and tasks.
3. DiffLM can generate various structured formats data.

**Weaknesses:**

1. Some very related works are missing. There are some works that also combine PLMs, VAE and diffusion. The authors should do a concrete survey.
2. The contribution is limited in my opinion. I think the main contribution: the authors align a encoder for LLM and do the experiments on the task, since doing generation in the latent space of VAE by diffusion model is quite common.

**Questions:**

1. For different data, do we need to train different VAE encoder? If yes, can we still call it as plug-and-play module?
2. As we know, diffusion models are good at generating diverse samples. So how to guarantee the generated samples are consistent? ( for exmaples, sometimes there will be some mismatches in the generated samples)
3. Besides training a VAE based on LLMs, what do you think is the major contribution of this work?
4. Besides specific data format, is it possible to use this method to do natural text generation? Will the generated text make sence?

---

> ### Author Response · Authors · 2024-11-24
> **Response to Reviewer rfz8**
>
> Thanks for your constructive comments and recognition of our work. Please see below for our responses to your concerns.
>
> **R1: Missing related work.**
>
> **A1:** We have conducted a more in-depth survey on LLMs with auto-encoders and plan to include more relevant references on language modeling task[1,2,3,4], unconditional text generation[5,6] and control text generation[7,8]. We would appreciate it if you could specify the important related works that we missed.
>
> **R2: Limited contribution.**
>
> **A2:** We respectfully disagree with the assessment that our contribution is limited. Our contributions and insights are as follows:
>
> **A2.1:** Motivation and Challenge: We pointed out the principles and challenges for high-quality structured text data synthesis, i.e., producing scalable and diverse synthetic data with reasonable relevant knowledge while maintaining the same data requirements (structure, topic, domain, etc.) as the target data. Existing prompt-based and fine-tuning methods cannot achieve both objectives at a low cost.
>
> **A2.2:** Framework: To address these goals, we propose the DiffLM framework, which decouples the task of learning the requirements of the data to be synthesized from the language modeling task. We model these requirements in the latent space and inject them into an unaltered LLM, enabling it to generate desired and realistic data. The synthetic data combines LLM's broad knowledge with specific data patterns learned from the training data, leading to enhanced performance on downstream tasks.
>
> **A2.3:** Techniques: For implementation, we propose a training recipe that keeps the LLM decoder fixed and only trains the VAE encoder and projector. We also validate different latent knowledge injection methods.
>
>
> Responses to Questions:
>
> **Q1:** [Role of VAE encoder] Do we need to train different VAE encoder? If yes, can we still call it as plug-and-play module?
>
> **A3:** Yes, for different data types, we train different VAE encoders. It is still plug-and-play because by swapping different VAE encoders and injectors, we can generate text data that meets the corresponding requirements without altering the generative model (LLM).
>
> **Q2:** How to guarantee the generated samples are consistent?
>
> **A4:** We designed simple post-processing methods to validate the structure of synthesized data (e.g., required columns for tables), and we found that only ~3% of the synthesized data has the wrong format, which demonstrates the consistency of the generated samples.
>
> **Q3:** Major contribution
>
> **A5:** As addressed in A2, our major contribution lies in proposing the DiffLM framework, which combines PLMs with diffusion techniques to synthesize data that merges the LLM's broad knowledge with specific data requirements learned from the training data.
>
> **Q4:** Is it possible to use this method to do natural text generation?
>
> **A6:** Yes. You can replace the training data with free text from a specific domain or topic, train DiffLM, and achieve generation of natural language in that specific domain or topic. You may refer to [9] for detailed data construction and training processes. Compared to [9], we have added latent diffusion techniques to address the discrepancy between the encoder's posterior and prior. Since our work focuses on synthesizing structural data, we did not include experiments in these scenarios due to space limitations and the main focus of the paper.
>
> Reference:
>
> [1] Lin et al. Text Generation with Diffusion Language Models: A Pre-training Approach with Continuous Paragraph Denoise. ICML 2023.
>
> [2] Yuan et al. Text Diffusion Model with Encoder-Decoder Transformers for Sequence-to-Sequence Generation. NAACL 2024.
>
> [3] Reid et al. DiffusER: Diffusion via Edit-based Reconstruction. ICLR 2023.
>
> [4] Zhang et al. DiffuSum: Generation Enhanced Extractive Summarization with Diffusion. ACL (Findings) 2023.
>
> [5] Yu et al. Latent Diffusion Energy-Based Model for Interpretable Text Modeling. ICML 2022.
>
> [6] Chen et al. A Cheaper and Better Diffusion Language Model with Soft-Masked Noise. EMNLP 2023.
>
> [7] He et al. DiffusionBERT: Improving Generative Masked Language Models with Diffusion Models. ACL 2023.
>
> [8] Austin et al. Structured Denoising Diffusion Models in Discrete State-Spaces. NeurIPS 2021.
>
> [9] Li et al. Optimus: Organizing Sentences via Pre-trained Modeling of a Latent Space. EMNLP 2020.

---

> > ### Comment · Reviewer_rfz8 · 2024-11-26
> >
> > Thanks for your clearification, it's clearer now. And I will raise my score to 6.

---

### Official Review · Reviewer_P77K · 2024-11-09

**Soundness:** 2
**Presentation:** 3
**Contribution:** 3
**Rating:** 5
**Confidence:** 5

**Summary:**

The paper presents DiffLM, a novel framework for synthetic data generation that combines LLMs with diffusion models. The primary goal is to address the challenges of generating high-quality synthetic data with a specific focus on structured formats such as tabular, code, and tool data. DiffLM introduces a plug-and-play latent feature injection module to decouple the learning of target distribution knowledge from the generative objectives of LLMs. The framework is evaluated on seven real-world datasets, demonstrating that DiffLM can outperform real data in certain downstream tasks.

**Strengths:**

1. The paper addresses the significant and relevant application of generative models: generating high-quality synthetic textual and structural data, which is an important field to study.
2. DiffLM is the first framework to combine diffusion models with LLMs for the purpose of high-quality data synthesis.

**Weaknesses:**

1. In Table 1, DiffLM does not consistently outperform TabSyn across various columns. While the authors compare state-of-the-art results based on language models, there are concerns:
- GReaT is based on GPT-2, which is a relatively weak baseline. A more robust baseline, such as GPT-4 or at least Mistral 7B, should be considered.
- Since TabSyn is also a diffusion-based table generation method, the integration of LLMs in DiffLM does not significantly enhance performance compared to TabSyn, suggesting that DiffLM may be a less effective solution for table generation.
2. The paper claims to focus on data generation with only a small set of real-world distribution data. However, for table and code data, the dataset size (~24k) is substantial. Consequently, specialized models like TabSyn have sufficient training data to perform well without LLM assistance, which diminishes the perceived advantage of DiffLM.
3. The study does not explore how the quantity of real & synthetic data affects downstream task performance respectively, which is crucial for understanding the practical impact of synthetic data.

In conclusion, certain details remain unclear in the paper (see questions), and some settings are not fully addressed.

**Questions:**

1. Is the injector module tuned in Figure 1? Clarity on this would be beneficial.
2. In Table 1, should the column "Beijing MLE" with a down arrow actually be an up arrow?
3. Which specific LLM is utilized in DiffLM presented in Table 1?
4. How many data samples were synthesized for each of the tasks evaluated?
5. What is the data volume used specifically for tool-related tasks?
6. How does the diversity of synthetic data impact downstream results? Can you control the generation diveristy of synthetic data?  An exploration of this aspect would be informative.

---

> ### Author Response · Authors · 2024-11-24
> **Response to Reviewer P77K**
>
> Thanks for your constructive comments. Please see below for our responses.
>
> **R1: Finetuning Baselines - GReaT.**
>
> **A1:** Yes, we attempted to validate the GReaT method on Mistral but found it could not directly and effectively generate data with the desired structure. GReaT organizes tabular data in a "key is value" format and uses a smaller PLM (i.e., GPT-2) for continued pretraining. However, when applied to larger models like Mistral, GReaT struggled to effectively generate the desired structured data. The sample generated by GReaT with Mistral is shown in follow. We hypothesize that controlling an LM by fine-tuning it with supervised data cause catastrophic forgetting for LLMs, as suggested by Luo et al[1]. Specifically, the "key is value" data constructed by the GReaT method, when used to continue pre-training Mistral, causes internal knowledge collapse - both undermining the model's existing knowledge and failing to do effective data synthesis. Additionally, training the adult dataset on GReaT for 200 epochs (default settings) requires approximately 50 hours on 8 A100 80G GPUs, which is resource-intensive. In contrast, DiffLM under the same training settings requires only about 7 hours.
>
> Synthetic data generated by GReaT with Mistral:
> ```
> <s> income is <=50K, $300/month  income is >50K, $600/month  I am trying to figure out how to implement the above logic with the following table I have:  | ID | Amt | |----|-----| | 1  | 450 | | 2  | 650 | | 3  | 500 | | 4  | 150 | | 5  | 700 | | 6  | 400 |  I have tried using `IF ELSE` function in Excel but I am not sure how to apply it to this particular situation.......</s>
> ```
>
> **R2: DiffLM does not consistently outperform TabSyn.**
>
> **A2:** We want to emphasize that our goal is a unified structured data synthesis framework that supports various domains like tabular data, codes, and tools, and tabular data generation in our work is just a subdomain of synthetic data generation. As a comparison, TabSyn is not applicable to more complex data synthesis tasks, such as code generation and tool usage, which involve generating longer content, more complex data types, and highly structured data, while our DiffLM can handle complicated scenarios. The results of tabular data synthesis are to demonstrate that our method possesses generality and can achieve on-par results with domain-specific models without being specifically tailored to a particular domain.
>
> **R3: How the quantity of real & synthetic data affects downstream task performance?**
>
> **A3:** We experimented with increasing the amount of data synthesized by DiffLM and combining real data with DiffLM-synthesized data. As shown in Table A, adding more synthesized data further improves around 0.2% MLE performance in the tabular scenario. Since our method can synthesize unlimited amounts of data and we did not design any complex post-processing method, the performance improvement brought by DiffLM-synthesized data in downstream tasks still has significant room for growth. Additionally, combining real and synthetic data generated by DiffLM can improve downstream performance; all results exceed >0.2% of those using only DiffLM data. Notably, on the Beijing and Shoppers datasets, the combination of real data and DiffLM synthetic data surpasses 0.6%-3% of the performance of training on real data alone.
>
>
> **Table A:** Tabular MLE performance with varying quantity of real and synthetic data. Performance on the Beijing dataset is evaluated using the RMSE metric, where lower values indicate better performance.
>
> | | Adult|Default|Magic|Shoppers|Beijing|
> |-----|-----|-----|-----|-----|-----|
> |Real|0.927|0.770|0.946|0.926|0.423|
> |TabSyn (SoTA)|0.915|0.764|0.938|0.920|0.582|
> |DiffLM (1x)|0.894|0.793|0.910|0.9122|0.717|
> |DiffLM (2x)|0.896 (+0.002)|0.795 (+0.002)|0.914 (+0.004)|0.9124 (+0.0002)|0.704 (-0.013)|
> |Real+DiffLM|0.925|**0.802**|0.936|**0.932**|0.494|
>
> Reference:
>
> [1] Luo et al. An Empirical Study of Catastrophic Forgetting in Large Language Models During Continual Fine-tuning. arXiv 2023.

---

> > ### Author Response · Authors · 2024-11-24
> > **Response to Questions**
> >
> > We split this due to reply length limitation. Response to Questions:
> >
> > **Q1:** Is the injector module tuned in Figure 1?
> >
> > **A4:** Yes, the injector module in Figure 1 is tunable; only the VAE Decoder (LLMs) is frozen. We will clarify this in the manuscript.
> >
> > **Q2:** Down arrow for "Beijing MLE".
> >
> > **A5:** No, the MLE column with a down arrow is correct because the Beijing dataset is a linear regression task, and we report the RMSE metric, whereas the lower values indicate better performance. We add necessary explanations to the paper.
> >
> > **Q3:** Which specific LLM is utilized in DiffLM?
> >
> > **A6:** We used Mistral-7B-Instruct-v0.3 in DiffLM.
> >
> > **Q4:** How many data samples were synthesized?
> >
> > **A7:** We synthesized the same number of data samples as the real training data for each dataset; details can be found in Table 5 of the paper.
> >
> > **Q5:** What is the data volume used for tool-related tasks?
> >
> > **A8:** We synthesized the same number of tools as in the ToolBench dataset, which is 16,464 tools.
> >
> > **Q6:** How does the diversity of synthetic data impact downstream results?
> >
> > **A9:** We used DCR (Distance to Closest Record) to evaluate the diversity of synthetic data. As shown in Figure 5, the data synthesized by DiffLM has a similar DCR distribution to the test data, demonstrating that the overall diversity of synthetic data is comparable to that of the real data.

---

> > > ### Comment · Reviewer_P77K · 2024-11-25
> > >
> > > Thank you for the clarification. I still have concerns that finetuning LLMs like Mistral may lead to knowledge collapse, but GPT-2 would not. There might be better settings to finetune LLMs. Additionally, using many demonstrations to prompt powerful LLMs such as GPT-4 is a common method for generating high-quality data. It is better to show comparisons in terms of cost and generation-quality. Finally, your method's unification or controllability "just" changes the training data, rather than providing true control for various tasks within a unified model. Overall, I will maintain my score.

---

> ### Author Response · Authors · 2024-11-25
> **Response to Reviewer P77K**
>
> We greatly appreciate the time and effort you have invested in our paper. Hope the following responses will help improve the paper's score.
>
> **1. Finetuning LLMs like Mistral may lead to knowledge collapse, but GPT-2 would not.**
>
> As noted in Luo et al.[1], "Our findings revealed that the CF (catastrophic forgetting) problem is generally prevalent in the continual fine-tuning of various LLMs. Moreover, as the model scale increases, LLMs exhibit a more severe degree of forgetting in domain knowledge, reasoning abilities, and reading comprehension skills." In our experiments, we similarly observed that larger LLMs (e.g., Mistral) often struggle with effective synthesis due to CF during the GReaT fine-tuning process. We agree with your point that better fine-tuning strategies could mitigate this issue. However, we believe our proposed DiffLM introduces a novel angle by leveraging diffusion techniques to effectively address challenges like knowledge collapse, providing a complementary alternative to existing methods such as GReaT.
>
> **2. MLE performance with GPT-4 prompting methods.**
>
> We understand your concerns regarding the use of few-shot prompting with GPT-4 for data synthesis. In Table B, we provided results where tabular data was synthesized using GPT-4 with 5-shot prompting approach. As the table shows, prompt-based methods consistently underperform compared to our DiffLM approach across all datasets. Specifically, in the Shoppers dataset, the prompt-based method lags behind DiffLM by 7.7%. While prompting methods may achieve adequate results in specific cases, they lack guarantees for generating diverse, large-scale synthetic datasets. We plan to include additional details about prompt designs and extend results across the other two datasets in the Appendix of the revised version.
>
> **Table B:** Tabular MLE performance with different synthetic data generation methods. Rho stands for column-wise distribution density estimation task, where lower values indicate better performance.
>
> | |**Adult MLE**|**Adult rho**|**Magic MLE**|**Magic rho**|**Shoppers MLE**|**Shoppers rho**|
> |:-----:|:-----:|:-----:|:-----:|:-----:|:-----:|:-----:|
> |Real|0.927|-|0.946|-|0.926|-|
> |GPT-Prompting|0.889|32.55|0.864|10.08|0.835|41.11|
> |GReaT|0.913|12.12|0.888|16.16|0.902|14.51|
> |DiffLM|0.894|9.16|0.910|7.04|0.912|14.43|
>
> **3. Controllability.**
>
> We would like to clarify that our primary goal is to propose a data synthesis strategy, rather than deliver a unified model. Specifically, our objective is to develop an effective framework for generating structured text data suitable for training downstream models. In this context, the produced data holds greater value than the model itself, so making a unified model is not a must-have in our task.
>
> Furthermore, our method does more than "just changes the training data". DiffLM innovatively integrates external data requirements into the generation process via a latent space without modifying the LLM’s parameters, thereby preserving LLM's general knowledge capabilities. Therefore, our method is not a trivial framework that only sets random values, but it leverages the LLM's rich knowledge to synthesize data that, in some downstream tasks, surpasses the performance of real data, which is a notable advancement over prior work.
>
> We acknowledge that our current approach does not support unified generation across diverse data structures, as it requires training both a denoising network and an injector network on domain-specific datasets. A potential solution is to introduce task-specific tokens, enabling unified task learning via a shared framework (e.g., training with task-specific soft prompts for different data synthesis tasks in a unified training process). Due to time constraints during this rebuttal phase, we will explore this direction in future work.
>
> We hope this addresses your concerns and clarifies the contributions of our work. Thank you again for your valuable feedback.
>
> Reference:
>
> [1] Luo et al. An Empirical Study of Catastrophic Forgetting in Large Language Models During Continual Fine-tuning. arXiv 2023.

---

> ### Author Response · Authors · 2024-11-28
>
> Dear reviewer P77K, considering the Rebuttal deadline is approaching, we sincerely hope to receive your response. If you have any questions or concerns regarding our explanation, please do not hesitate to contact us. Your response is of great significance to us for improving this work, and we look forward to hearing from you.

---

### Official Review · Reviewer_hoMx · 2024-11-09

**Soundness:** 3
**Presentation:** 2
**Contribution:** 2
**Rating:** 5
**Confidence:** 5

**Summary:**

This paper introduces DiffLM, a novel framework for controllable synthetic data generation using large language models (LLMs). DiffLM addresses the challenges of LLM-based data synthesis, such as limited understanding of target data distributions and complex prompt engineering, particularly for structured data. The key innovation lies in decoupling the learning of data distributions from the LLM's training objectives. This is achieved by employing a variational autoencoder (VAE) to learn a latent representation of the real data, which is then used to guide the LLM's generation process. To enhance the quality of the learned latent space, a diffusion model is incorporated, mitigating the limitations of traditional VAEs in capturing complex distributions. Furthermore, a soft prompt injection module seamlessly integrates the learned latent information into the LLM decoding process without retraining, preserving the LLM's inherent knowledge and reasoning abilities. The authors evaluate DiffLM on seven real-world structured datasets spanning tabular, code, and tool data. Their experiments demonstrate that DiffLM generates high-quality synthetic data, achieving comparable or even superior performance to existing methods on downstream tasks, and even surpassing the performance of real data in certain cases. The proposed framework offers a flexible and robust approach for controllable data synthesis, paving the way for wider adoption of LLMs in various data generation scenarios.

**Strengths:**

- DiffLM introduces a combination of VAEs, diffusion models, and LLMs for synthetic data generation. The incorporation of a latent diffusion model within the VAE framework, coupled with the soft-prompt injection mechanism, distinguishes DiffLM from prior work that primarily focuses on either fine-tuning LLMs or using simpler latent variable models for text generation.
- The paper provides a thorough evaluation across a diverse set of structured datasets and tasks (tabular, code, and tool generation), demonstrating the robustness and adaptability of the proposed framework.
- The observation that synthetic data generated by DiffLM can outperform real data on certain downstream tasks is  compelling and suggests a potential for knowledge enhancement through synthetic data.
- The proposed DiffLM framework addresses a significant challenge in leveraging LLMs for data synthesis, namely, controlling the generated data's structure and distribution. The ability to generate high-quality synthetic data for structured formats has broad implications for various applications, including data augmentation, privacy preservation, and software testing.

**Weaknesses:**

- This work suggests that the proposed approach can achieve controllable synthetic data generation, but the model didnt use any disentangled latent variables learning or other grounding techniques to achieve such controllability, can the authors clarify what exactly the controllability refers to?
- In experiments section the works shows that the synthetic data can outperform real data for continued pre-training, which is intriguing and interesting, it would be more helpful to provide more analysis and empirical insights on why and how this is made possible.
- How does the interpolation of latent space look like? How and where did the diffusion model improve the issues of regular VAE model as described in the main text?
- While automated metrics like DCR and downstream task performance are useful, they do not fully capture the nuances of data quality, especially for complex structured data. Incorporating human evaluation, particularly for code and tool generation, would provide a more holistic assessment of the generated data's usability and realism.

**Questions:**

Please see comments above

---

> ### Author Response · Authors · 2024-11-24
> **Response to Reviewer hoMx**
>
> Thank you for your valuable comments. We appreciate Reviewer hoMx's recognition of our work on DiffLM for high-quality data synthesis. We address your concerns below:
>
> **R1: [Controllability Clarification] Can the authors clarify what exactly the controllability refers to?**
>
> **A1:** We define the controllability of text data synthesis as the ability to generate text that satisfies desired requirements (e.g., structure, topics, domains)[1,2]. Existing methods for structured textual data synthesis often struggle with controllability. On one hand, LLM prompt-based methods relying on prompt engineering or few-shot inference cannot guarantee the diversity and scalability of synthetic data, even with complex human-crafted processes[3]. On the other hand, controlling a LM by fine-tuning it with supervised data (SFT, RLHF) is not only expensive but might also degrade the LLM's general capability[1,4]. Our method addresses these challenges through sampling in the latent space while maintaining data structure due to LLM's instruction-following ability. Thanks for pointing this out, we will update related parts in the next manuscript.
>
> **R2: [Analysis on Synthetic Data Outperforming Real Data]  It would be more helpful to provide more analysis and empirical insights on why and how this (the synthetic data can outperform real data for continued pre-training) is made possible.**
>
> **A2:** Our motivation arises from observing that many works attempt to use LLMs for data synthesis but often face difficulties in efficiently generating desired and realistic high-quality data[3] at scale. We propose DiffLM, a framework designed to steer LLMs in data generation by decoupling the task of capturing the characteristics of the target dataset from the core language modeling process. We model the nature of the target dataset in the latent space and then inject it into the unaltered and carefully aligned LLM, enabling it to generate desired and realistic data. As shown in Figure 2 of our paper, DiffLM synthesizes data that leverages the learned target data latent representation and the LLM’s internal knowledge, resulting in better judgment scores than the real data. We believe this is because the synthetic data combines LLM's broad knowledge with specific data patterns learned from the vast quantity of training data, leading to enhanced performance on downstream tasks.
>
> **R3: Interpolation of Latent Space and Role of Diffusion Model**
>
> **A3.1:** In Figure 6, we provide t-SNE visualizations of the latent space learned by DiffLM across multiple datasets. The latent space exhibits clear clustering and hierarchical structures. Specifically, for the Beijing dataset, samples with larger numerical values are distributed in the upper part of the 2D t-SNE space, while those with smaller values are in the lower part, forming a gradient from top to bottom.
>
> **A3.2:** Regarding the role of the latent diffusion model, in our early experiments using only a VAE to learn the latent space, only about 10% of the synthetic samples had structures consistent with the training data (e.g., in tabular data, the synthetic data included all the required columns). The latent diffusion model addressed this sampling failure problem, increasing the pass rate to about 97%. As mentioned in our reply to Reviewer rfz8, Question 3, we will include the relevant information in the next version.
>
> **R4: Human Evaluation**
>
> **A4:** We agree that automated metrics like DCR and downstream task performance may not fully capture the nuances of data quality for complex structured data. In fact, we have used GPT-4 to rate and perform preference judgments on synthesized tools and real tools. The results in Figure 2 and Table 3 demonstrate the quality of our synthesized data. As per your suggestion, we have conducted human evaluations on the tool usage data. Specifically, we compared 100 pairs of randomly selected DiffLM-generated data and real data within the same category. As shown in Table A, our synthetic data is preferred by human annotators.
>
> **Table A: Human Evaluation**
> |  | Percentage |
> |-------------------|------------|
> | DiffLM Win        | 88%        |
> | Equal             | 6%         |
> | Real Win          | 6%         |
>
> **Reference:**
>
> [1] Keskar et al. CTRL: A Conditional Transformer Language Model for Controllable Generation. arXiv 2019.
>
> [2] Li et al. Diffusion-LM Improves Controllable Text Generation. NeurIPS 2022.
>
> [3] Long et al. On LLMs-Driven Synthetic Data Generation, Curation, and Evaluation: A Survey. ACL 2024.
>
> [4] Borisov et al. Language Models are Realistic Tabular Data Generators. ICLR 2023.

---

> ### Author Response · Authors · 2024-11-28
>
> Dear reviewer hoMx, considering the Rebuttal deadline is approaching, we sincerely hope to receive your response. If you have any questions or concerns regarding our explanation, please do not hesitate to contact us. Your response is of great significance to us for improving this work, and we look forward to hearing from you.

---

### Author Response · Authors · 2024-12-03
**General Response**

Dear Program Chairs, Area Chairs, and Reviewers,

First, we sincerely thank you for your valuable time, constructive feedback, and insightful guidance. We are delighted that the reviewers recognize the novelty and effectiveness of our proposed methods, as well as our exploratory efforts in high-quality text data synthesis. At the same time, we acknowledge the concerns raised, primarily regarding the contributions and scope of our work, and we provide a unified explanation below.

Following prior work[1,2], we define the controllability of text data synthesis as the ability to generate text that satisfies specific requirements (e.g., structure, topics, domains). Existing methods for structured textual data synthesis often face challenges in achieving this controllability. On one hand, LLM prompt-based methods that rely on prompt engineering or few-shot inference cannot ensure the diversity and scalability of synthetic data, even with intricate human-crafted designs[3]. On the other hand, controlling a language model (LM) by fine-tuning it with supervised data (SFT, RLHF) is not only computationally expensive but may also degrade the LM's general capabilities[1,4]. Our proposed method addresses these challenges by sampling in the latent space while leveraging the LLM's instruction-following abilities to preserve data structure.

Regarding contributions, our work makes the following key advancements:

- Motivation and Challenges: We identify the principles and challenges for high-quality structured text data synthesis, emphasizing the need for scalable and diverse synthetic data that integrates relevant knowledge while satisfying specific data requirements (structure, topic, domain). Current prompt-based and fine-tuning methods struggle to meet these objectives cost-effectively.

- Framework: To address these challenges, we propose the DiffLM framework, which decouples the task of learning the synthesis requirements from language modeling. By modeling these requirements in the latent space and injecting them into an unaltered LLM, our framework generates realistic data that combines the LLM’s extensive knowledge with the specific patterns learned from training data. This leads to superior performance on downstream tasks.

- Techniques: We introduce a training procedure that keeps the LLM decoder fixed, training only the VAE encoder and projector. Furthermore, we validate various methods of latent knowledge injection to ensure effectiveness.

Additionally, in response to the reviewers' suggestions, we have: 1) Enhanced our explanations of baseline training parameters. 2) Clarified how the Latent Diffusion model contributes to DiffLM's improvements. 3) Addressed other potentially ambiguous descriptions.

More comprehensive analyses, including those on the quantity of synthetic data, synthetic data outperforming real data, and human evaluation, have been provided in the rebuttal responses and the revised paper.

We also plan to open-source all intermediate data and code following the review process. We sincerely look forward to further discussions and appreciate your thoughtful consideration of our work.

Best regards,

The author(s) of Paper 10707

Reference:

[1] Keskar et al. CTRL: A Conditional Transformer Language Model for Controllable Generation. arXiv 2019.

[2] Li et al. Diffusion-LM Improves Controllable Text Generation. NeurIPS 2022.

[3] Long et al. On LLMs-Driven Synthetic Data Generation, Curation, and Evaluation: A Survey. ACL 2024.

[4] Borisov et al. Language Models are Realistic Tabular Data Generators. ICLR 2023.

---

### Meta-Review · Area_Chair_BC7c · 2024-12-18

**Metareview:**

This paper presents a framework that combines large language models with diffusion models for synthetic data generation, particularly for structured data like tables, code, and tools. The key innovation lies in using a VAE with latent diffusion to capture data distributions and inject them into LLMs via soft prompts, without model retraining. The strengths include competitive performance across diverse structured data types, with synthetic data sometimes outperforming real data on downstream tasks, and a novel approach to maintaining LLM capabilities while controlling generation. However, several critical weaknesses emerge: (1) insufficient justification for choosing diffusion over simpler alternatives like expressive priors, (2) limited ablation studies on key architectural choices, (3) inadequate comparison with strong baselines using modern LLMs, (4) unclear evidence supporting claims about VAE representation being ignored by LLMs, and (5) some key relevant reference regarding latent discrete diffusion is missing, including Latent Diffusion for Language Generation (Lovelace et al. 2022) and PLANNER: Generating Diversified Paragraph via Latent Language Diffusion Model (Zhang et al. 2023). The recommendation to reject stems primarily from these methodological gaps that raise questions about the true advantages of the proposed approach over existing methods.

**Additional Comments On Reviewer Discussion:**

The author response sparked some discussion around three main points: the failure of fine-tuning larger LLMs, the choice of diffusion models, and ablation studies. The authors provided examples showing how fine-tuning Mistral with the GReaT approach led to poor generation, and presented additional ablation results comparing different denoising network architectures. While one reviewer (rfz8) was satisfied with the clarifications and raised their score to 6, others maintained their original scores of 5, expressing continued concerns about the justification for using diffusion models and the limited exploration of alternatives. The authors' responses, while detailed, did not fully address the fundamental concerns about methodological choices and baseline comparisons. The consistent skepticism from multiple reviewers about core design decisions, combined with the limited evidence supporting key claims, still remains.

---

### Decision · Program_Chairs · 2025-01-22

Reject